# Renoprotective Effects of Daprodustat in Patients with Chronic Kidney Disease and Renal Anemia

**DOI:** 10.3390/ijms25179468

**Published:** 2024-08-30

**Authors:** Yoshitaka Shimada, Yuichiro Izumi, Yukiko Yasuoka, Tomomi Oshima, Yasushi Nagaba, Masayoshi Nanami, Jeff M. Sands, Noriko Takahashi, Katsumasa Kawahara, Hiroshi Nonoguchi

**Affiliations:** 1Division of Internal Medicine, Kitasato University Medical Center, 6-100 Arai, Kitamoto 364-8501, Saitama, Japan; simada-h@insti.kitasato-u.ac.jp (Y.S.); nagaba-y@insti.kitasato-u.ac.jp (Y.N.); 2Department of Nephrology, Kumamoto University Graduate School of Medical Sciences, 1-1-1 Honjo, Chuo-ku, Kumamoto 860-8556, Kumamoto, Japan; izumi_yu@kumamoto-u.ac.jp; 3Department of Physiology, Kitasato University School of Medicine, 1-15-1 Kitasato, Minami-ku, Sagamihara 252-0374, Kanagawa, Japan; yasuoka@med.kitasato-u.ac.jp (Y.Y.); tomomio@kitasato-u.ac.jp (T.O.); ntakahas@med.kitasato-u.ac.jp (N.T.); kawahara@kitasato-u.ac.jp (K.K.); 4Department of Cardiovascular and Renal Medicine, Hyogo Medical University, 1-1 Mukogawa-cho, Nishinomiya 663-8501, Hyogo, Japan; m-nanami@hyo-med.ac.jp; 5Renal Division, Department of Medicine, Emory University School of Medicine, 1639 Pierce Drive, WMB Room 1107, Atlanta, GA 30322, USA; jeff.sands@emory.edu; 6Nephrology, Internal Medicine, Sagamihara Red-Cross Hospital, 256 Nakano, Midori-ku, Sagamihara 252-0157, Kanagawa, Japan

**Keywords:** erythropoietin, PHD inhibitor, hypoxia, deglycosylation, proximal tubules, distal tubules, renal erythropoietin-producing interstitial cells

## Abstract

Many large-scale studies revealed that exogenous erythropoietin, erythropoiesis-stimulating agents, have no renoprotective effects. We reported the renoprotective effects of endogenous erythropoietin production on renal function in ischemic reperfusion injury (IRI) of the kidney using the prolyl hydroxylase domain (PHD) inhibitor, Roxadustat. The purpose of this study was to investigate the effects of daprodustat on the progression of chronic renal failure. We retrospectively investigated the effects of daprodustat on the progression of chronic renal failure and renal anemia in patients with stages 3a-5 chronic kidney diseases (estimated glomerular filtration rate, eGFR < 60 mL/min/1.73 m^2^). The results show that daprodustat largely slowed the reduction in eGFR. The recovery of renal function was observed in some patients. Daprodustat is useful not only for renal anemia but also for the preservation of renal function. The renoprotective effect of daprodustat was small in patients with serum creatinine larger than 3–4 mg/dL because of low residual renal function. The appearance of renal anemia would be a sign of the time to start using daprodustat.

## 1. Introduction

Erythropoietin (Epo) was found and purified from urine of anemic patients [1,2]. Erythropoiesis-stimulating agents (ESAs) and hypoxia-inducible factor-prolyl hydroxylase (HIF-PHD) inhibitors have largely changed the treatment of renal anemia [3,4,5,6,7,8,9]. The appearance of ESAs created the expectation of renoprotective effects of ESAs. However, large scale studies showed the lack of renoprotective effects of ESAs on the progression of renal failure in patients with chronic kidney disease (CKD) [10,11,12,13]. We showed the renoprotective effects of roxadustat, a HIF-PHD inhibitor, in rats with ischemic reperfusion injury (IRI) of the kidney [14]. Roxadustat induced Epo production not only by interstitial cells as ESAs but also by the proximal and distal tubules. A small amount of Epo production by the nephron was enough to cause the renoprotective effect in the IRI model. Although the renoprotective effects of HIF-PHD inhibitors in acute kidney injury (AKI) have been reported, the renoprotective effects in chronic renal failure have not been reported. The effect of daprodustat and vadadustat on renal anemia was compared with ESA [15,16]. The absence or presence of the renoprotective effects of daprodustat or vadadustat was not examined in these trials. The failure of showing renoprotective effects through large scale studies with ESAs and the same stimulation of interstitial cells-derived Epo production both by ESAs and HIF-PHD inhibitors have counteracted the examination of the possibility of renoprotection by HIF-PHD inhibitors [4,7,8,17,18,19]. The purpose of this study was to examine whether daprodustat has renoprotective effects on the progression of chronic kidney diseases.

We developed the detection of Epo by deglycosylation-coupled Western blotting [20,21,22]. We showed the presence of tubular production of Epo by roxadustat, which induced renoprotective effects in an acute kidney injury (AKI) model [14]. Although the mechanisms of improvement of renal anemia are the same with ESAs and HIF-PHD inhibitors, ESAs are the end-product of HIF stimulation, while HIF-PHD inhibitors stimulate much earlier steps of producing Epo. Therefore, HIF-PHD inhibitors have the possibility of some additional effects compared with ESAs [23]. We examined the effects of daprodustat on the progression of chronic renal failure in patients with stages 3a-5 CKD and renal anemia (eGFR < 60 mL/min/1.73 m^2^).

## 2. Results

### 2.1. Effects on Anemia

After an observation period of 43.2 ± 29.8 (mean ± SD) months, daprodustat (2 or 4 mg/day in nineteen and four patients, respectively) was administered for 15.9 ± 8.4 months. Daprodustat significantly increased the plasma hemoglobin level from 9.9 ± 1.1 to 11.0 ± 1.1 * g/dL (* *p* < 0.005 by paired *t*-test).

### 2.2. Effects on Renal Function

Figure 1 shows three typical cases of the change in the slope of eGFR (estimated glomerular filtration rate) decline [24,25,26] by daprodustat. Daprodustat decreased the slope of eGFR decline from −0.42 ± 0.47 to 0.00 ± 0.34 mL/min/1.73 m^2^/month (Figure 2, * *p* < 0.001 by Wilcoxon signed-rank test). A total of 20 out of 23 patients showed the reduction of the decrease in eGFR (slopes of the decrease in eGFR).

### 2.3. Effects on Urinary Protein Excretion

Daprodustat did not change urinary protein excretion (1.22 ± 1.36 to 0.95 ± 0.81 g/creatinine, *p* > 0.05 by Wilcoxon signed-rank test).

### 2.4. Interaction of Renoprotective Effects and Serum Creatinine and Hemoglobin Levels

Whether the renoprotective effects of daprodustat is influenced by eGFR, serum creatinine or hemoglobin levels at the starting point were investigated. Figure 3 and Figure 4 show that the renoprotective effect of daprodustat was not observed in patients with eGFR less than 10 mL/min/1.73 m^2^, or serum creatinine levels more than 4 mg/dL, respectively. Renoprotective effects were seen in patients with serum creatinine levels less than 3.

Plasma hemoglobin levels did not affect the renoprotective effects of daprodustat (Figure 5).

### 2.5. Interaction of Renoprotective Effects (The Changes in the Slopes of the eGFR) and the Increase in Hemoglobin Level (ΔHb)

In our study, the renoprotective effect of daprodustat was mildly related with erythropoietic effects (Figure 6). Therefore, increasing the dose of daprodustat may be considered when renoprotective effects are not observed. The renoprotective effects of daprodustat was corelated with the the decrease in urinary protein excretion (Figure 7).

### 2.6. Multiple Regression Analysis

The factors influencing the change in eGFR were analyzed using multiple regression analysis with forward Stepwise selection. The object variable was the change in the slope of eGFR. The explanatory variables are age, eGFR at the start, s-Cr at the start, pre-Hb, post-Hb, delta Hb, pre-U-TP/Ucr, post-U-TP/Ucr, delta U-TP. Multiple comparison analysis revealed that the adjusted coefficient of determination (R square) was 0.343 and 0.410 for delta U-TP and delta U-TP + delta Hb, respectively. These results are same with the findings in Figure 6 and Figure 7.

## 3. Discussion

These data show that daprodustat improved the decline of renal function in patients with stage 3a-5 CKD and renal anemia. We showed that roxadustat stimulates Epo production not only by the interstitial cells but also by the proximal tubules [14,27]. ESAs and HIF-PHD inhibitors improve renal anemia by stimulating Epo production by the interstitial cells [3,4,5,6,7,8,9,18]. HIF-PHD inhibitors, but not ESAs, have renoprotective effects not by correcting anemia but probably by stimulating tubular Epo production. HIF-PHD inhibitors will largely change the treatment of CKD patients with chronic renal failure and renal anemia as a first-choice drug. The appearance of anemia is caused by the decline of renal function, and it would be the sign to start to use HIF-PHD inhibitors. The elongation of the time until the induction of renal replacement therapy by the use of HIF-PHD inhibitors will provide a large benefit for CDK patients.

One of the striking findings of this study is the improvement of renal function that was observed in some patients. The decrease in renal function has been considered to be irreversible. Since roxadustat stimulates Epo production by the nephron, an improvement of renal tubular function could be possible. Multiple regression analysis with Stepwise selection revealed the decrease in urinary protein excretion to be the most explanatory factor. Tubular reabsorption of urinary protein or the improvement of glomerular function could be possible. Multiple regression analysis also showed the interaction between the renoprotective effect and the improvement of renal anemia. Renoprotective effects of daprodustat have some correlation with the decrease in urinary protein excretion and the improvement of renal anemia.

Figure 3 and Figure 4 show that the renoprotective effect of daprodustat is low in patients with high serum creatinine or low eGFR. It is because of the low residual renal function in those patients. The renoprotective effect of daprodustat was observed even in patients with severely reduced renal function. Patients with a serum creatinine less than 3 mg/dL or eGFR larger than 20 mL/min/1.73 m^2^ would have a better chance for the preservation of renal function by starting daprodustat.

Recently, sodium-glucose cotransporter 2 (SGLT2) inhibitors have also been shown to have effects on renal anemia [28]. The increase in plasma Epo concentration by SGLT2 inhibitors is very small, suggesting the lack of Epo production by interstitial cells. A small increase in plasma Epo concentration may suggest the possibility of Epo production by the nephron, same as with HIF-PHD inhibitors. The renin-angiotensin-aldosterone system is known to stimulate Epo production [29,30,31,32,33,34]. We showed that fludrocortisone stimulates Epo production by the nephron. These data suggest that Epo production by the nephron regulates renal function by affecting the renin-angiotensin-aldosterone system.

Epo producing nephrons are not so many. We showed the lack of autophagy or apoptosis by roxadustat in the IRI model AKI [14]. However, the small population of Epo-producing nephrons in the kidney cannot deny the possibility of the presence of autophagy or apoptosis from using roxadustat. Current Western blotting using renal cortex cannot detect such small changes in the nephron. Immunohistochemistry is the most sensitive tool to detect Epo production by the nephron. The precise mechanisms of Epo production by the nephron have to be examined by measuring single cell levels.

Our study has a large limitation because it is a retrospective study, and a small number of patients were enrolled. Therefore, future independent prospective studies that enroll a large number of patients are required to confirm our hypothesis.

In summary, daprodustat improved the decline of renal function in patients with CKD and renal anemia, probably by stimulating Epo production by the nephron.

## 4. Materials and Methods

### 4.1. Patients

We retrospectively examined the effects of daprodustat on the progression of CKD. Patients with stages 3a-5 (eGFR < 60 mL/min/1.73 m^2^) CKD and with renal anemia (plasma hemoglobin less than 13 and 12 g/dL in males and females, respectively) were enrolled in this study. Most patients have hypertension and received antihypertensive drugs (angiotensin II receptor blockers 17, calcium channel blockers 15 and diuretics 9 patients). Nine patients received a statin to treat hyperlipidemia. (Table 1). Patients who received renal replacement therapy were excluded. Patients taking steroid or immunosuppressive drugs were excluded. Patients taking diuretics or mineralocorticoid receptor antagonists were enrolled unless the doses were not changed during the study period. 

A total of 23 patients with CKD and renal anemia (chronic glomerulonephritis 9; benign nephrosclerosis 11, diabetic nephropathy 3, female 13; male 10) were enrolled in this study Table 2). Median age of the patients was 79.2 (female; median 82.6; IQR 75.0–88.0; male, median 76.0; IQR 67.5–84.5 y.o.). Height was 147 ± 8.3 (mean ± SD) and 160.1 ± 6.7 cm in females and males, respectively. Body weight was 45.5 ± 9.7 and 57.7 ± 12.8 kg in female and male patients, respectively. Mean BMI was 21.5 ± 3.4. An amount of 2 or 4 mg daprodustat was daily given to the all patients.

Blood chemistry was examined to measure renal function (serum creatinine (Cr) and eGFR) and plasma hemoglobin level at intervals of 1–6 months. Urinary protein excretion was also examined by measuring urinary protein excretion/urinary creatinine excretion (U-TP/Ucr). Serum creatinine levels were from 0.79 to 7.11 mg/dL (median 1.93, IQR 1.28–2.39 mg/dL). eGFR levels were from 7.0 to 52.7 mg/dL (median 24.3 IQR 16.7–33.2 mL/min/1.73 m^2^).

Since our study is a retrospective one, informed consent was obtained after the start of the investigation.

### 4.2. Renal Function Measurements

All patients received daprodustat and the changes in renal function were examined by the slope of decline of eGFR between pre- and post-treatments [24,25,26]. eGFR in Japan was calculated by eGFR (male) = 194 × Cr^−1.094^ × Age^−0.287^ and eGFR (female) = 0.739 × eGFR (male) [35,36,37].

### 4.3. Statistical Analyses

Data are expressed as mean ± SD. Statistical analyses were performed using Excel Statics (BellCurve, Tokyo, Japan). Statistical significance between two groups was analyzed using paired *t*-test or Wilcoxon signed-rank test after checking the normal distribution by Kolmogorov and Shapiro–Wilk test. To examine the participation of explanatory factors to the objective variable, multiple regression analysis with forward Stepwise selection was used. *p* < 0.05 was considered statistically significant. 

## Figures and Tables

**Figure 1 ijms-25-09468-f001:**
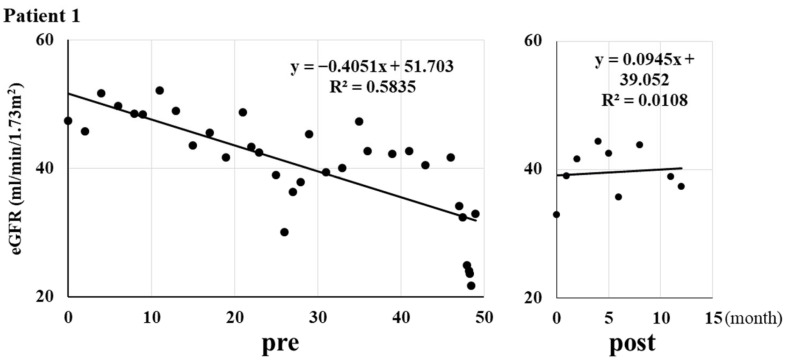
The changes in the eGFR slope in three patients before and after (pre and post, respectively) the administration of daprodustat. The line was drawn using least square method (Excel 2019).

**Figure 2 ijms-25-09468-f002:**
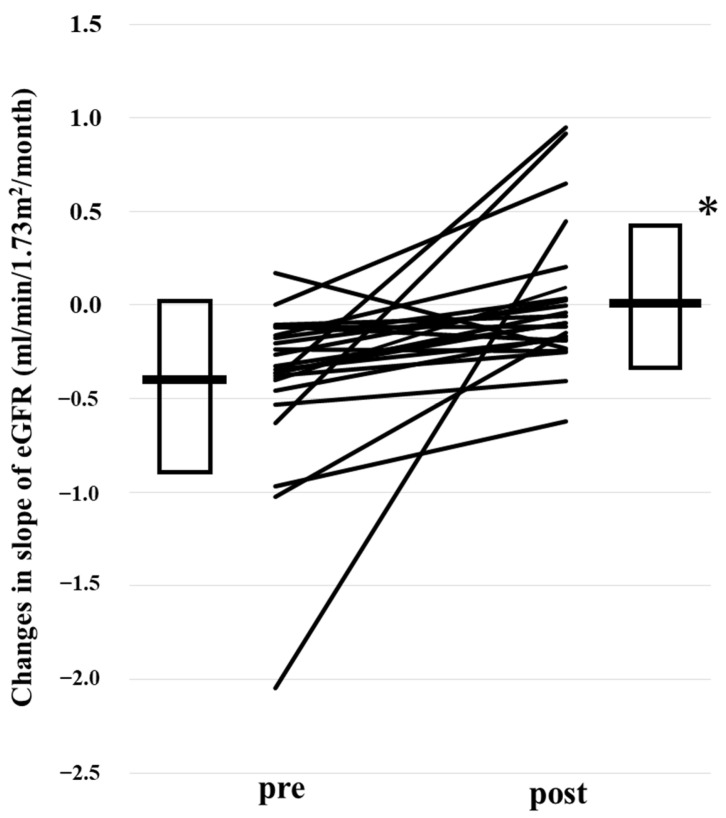
The change in the slope of the decline of eGFR (mL/min/1.73 m^2^/month) by daprodustat. Daprodustat significantly reduced the decline of eGFR slope. Box and bar show mean + SD.

**Figure 3 ijms-25-09468-f003:**
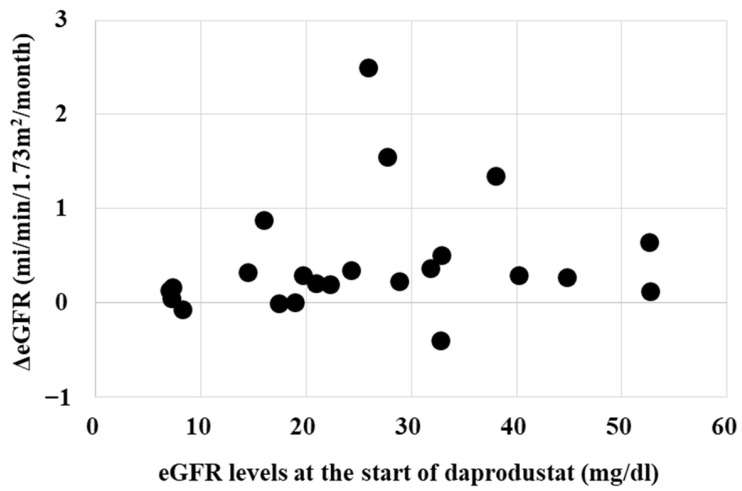
Relationship between eGFR levels at the start point and changes in eGFR. Renoprotective effects of daprodustat (reduction of the loss of eGFR) was not evident in patients with eGFR levels at the start point less than 10 mL/min/1.73 m^2^. ΔeGFR, the changes in the slopes of the decline of eGFR.

**Figure 4 ijms-25-09468-f004:**
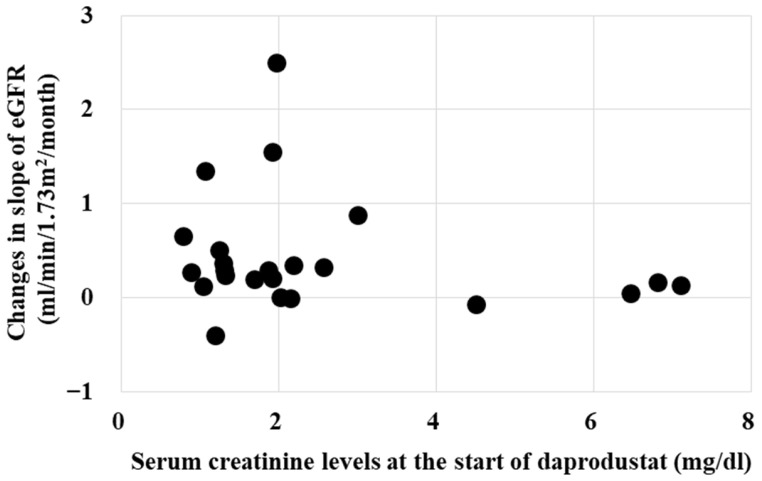
Relationship between serum creatinine levels and changes in eGFR. Renoprotective effect of daprodustat (reduction in the slopes of the decrease in eGFR) was small in patients with serum creatinine levels more than 4 mg/dL. Renoprotective effect was seen mainly in patients with serum creatinine levels less than 3 mg/dL.

**Figure 5 ijms-25-09468-f005:**
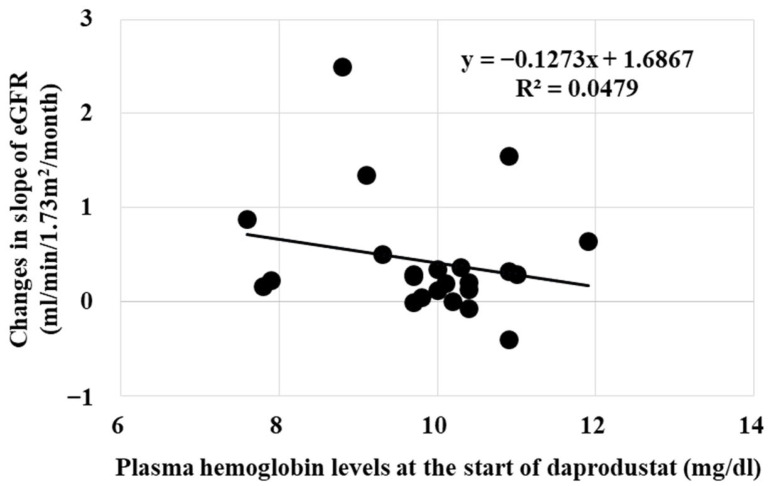
Relationship between plasma hemoglobin levels and the changes in eGFR. Renoprotective effects of daprodustat did not correlate with plasma hemoglobin levels.

**Figure 6 ijms-25-09468-f006:**
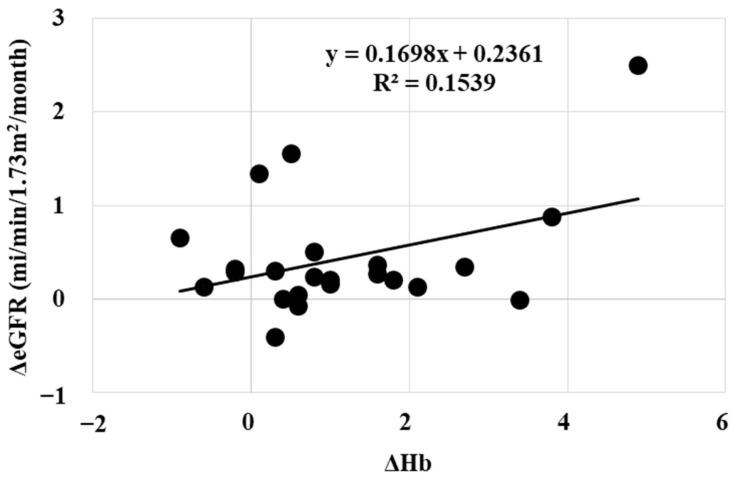
Relationship between the changes in hemoglobin levels (ΔHb) and the changes in eGFR. Renoprotective effects of daprodustat did not correlate with plasma hemoglobin levels.

**Figure 7 ijms-25-09468-f007:**
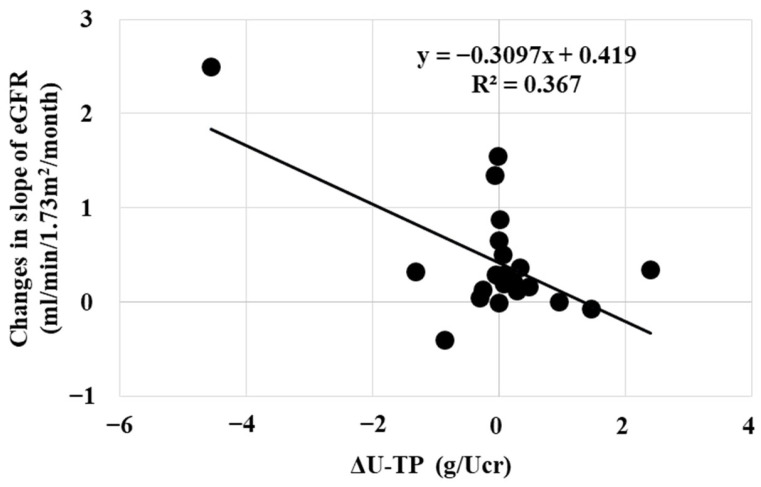
Interaction between the changes in slope of eGFR and the changes in urinary protein excretion (ΔU-TP). The reduction in the decrease in the slope of eGFR was correlated with the change in urinary protein excretion as shown by the multiple regression analysis.

**Table 1 ijms-25-09468-t001:** Basic characteristics of the patients and complications.

Sex	No.	Age	Height	BW (kg)	BMI	s-Cr	eGFR	Hypertens	Hyperlipi
female	13	82	147.4	45.5	20.8	1.74	28.4	85%	31%
SD		9	8.3	9.7	3.0	0.98	13.3		
male	10	76	160.1	57.7	22.4	3.38	22.9	80%	50%
SD		9.4	6.7	12.8	3.8	2.42	14.9		
total	23	79.2	152.9	50.8	21.5	2.45	26.0	83%	39%
		9.4	9.9	12.5	3.4	1.9	14.0		

s-CR, serum creatinine (mg/dL); Hypertens, hypertension; Hyperlipi, hyperlipidemia.

**Table 2 ijms-25-09468-t002:** Patients’ profile at the start of daprodustat and changes by the treatment.

Age	Sex	Pri Dis	s-Cr	eGFR	Pre-Period	Post-Period	Pre-Hb	Post-Hb	Pre-UTp/Ucr	Post-Utp/Ucr
70	f	NScl	1.3	33.5	125	9	10.3	11.9	1.27	1.610
72	m	CGN	1.93	27.7	52	11	10.9	11.4	0.26	0.28
73	f	CGN	1.25	32.9	49	12	9.3	10.3	1.27	1.33
92	f	CGN	0.89	49.6	15	23	9.7	11.3	ND	0.31
87	m	NScl	1.31	40	32	20	11.0	10.8	0.13	0.09
91	f	NScl	1.33	28.9	89	13	7.9	8.7	1.68	1.80
65	f	CGN	4.51	8.3	60	5	10.0	11.5	1.37	1.80
82	f	NScl	0.79	52.6	43	9	11.9	11.0	0.00	0.00
83	m	NScl	1.04	52.7	37	26	9.7	10.2	0.17	0.46
94	f	NScl	1.87	19.3	102	12	10.4	11.0	0.34	0.25
62	m	DMN	6.82	7.3	39	2	7.8	8.8	ND	4.13
88	f	NScl	1.7	22.3	20	30	10.1	11.1	0	0.08
66	m	CGN	1.93	21.0	38	16	10.4	10.5	0.54	1.08
75	f	NScl	2.02	19.0	44	27	10.2	11.1	1.62	2.76
66	m	CGN	7.11	7.0	21	17	10.4	10.5	2.79	2.12
86	m	NScl	3.01	16.0	15	21	7.6	11.4	0.07	0.08
85	m	DMN	1.98	25.9	38	10	8.8	13.7	5.51	0.95
72	m	DMN	2.19	24.3	8	20	10.0	12.7	0.37	2.77
82	f	CGN	2.58	14.5	16	31	10.9	10.7	2.11	0.80
80	m	NScl	6.48	7.2	31	19	9.8	10.4	0.91	0.61
87	f	CGN	1.20	32.8	33	29	10.9	11.4	1.94	1.08
85	f	CGN	2.15	17.4	51	5	9.1	13.1	ND	ND
78	f	NScl	1.0P7	38.0	66	2	9.1	9.2	0.2	0.13

(Pri-Dis, primary disease of CRF; s-Cr, serum creatinine (mg/dL); ND, not determined).

## Data Availability

The data presented in this study are openly available.

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
