# Peer review of "Renoprotective Effects of Daprodustat in Patients with Chronic Kidney Disease and Renal Anemia"

_ijms, 2024, doi:10.3390/ijms25179468_

Round 1

Reviewer 1 Report

Comments and Suggestions for Authors

In this study, the authors examined the renoprotective effects of daprodustat on CKD patients. By comparing to their pre-treatment renal function, the authors found that daprodustat treatment helped delay the eGFR decline in most patients. This information is clinically valuable, though there are some concerns about the study. 

1. For patient enrollment, the inclusion and exclusion criteria are not clear. In the 23 patients they enrolled, there is a significant variation in the post-daprodustat treatment period, which may cause a large variation of the reliability when they examine the slope of eGFR decline. 

2. More detailed information about the patients is needed. For example, what is the dosage and frequency of the daprodustat treatment of these patients? In addition to hypertension and hyperlipidemia, do they have any other health conditions that may potentially affect renal function, e.g. diabetes? Did they receive any other treatment such as dialysis? Their ethnicity? The authors may need to follow appropriate report guidelines, e.g. STROME. 

3. The limitations of the study need to be discussed. For example, this study only enrolled 23 patients. The small sample size may limit the reliability of the study. 

Comments on the Quality of English Language

This manuscript is well-written in English. 

Author Response

Comments 1: For patient enrollment, the inclusion and exclusion criteria are not clear. In the 23 patients they enrolled, there is a significant variation in the post-daprodustat treatment period, which may cause a large variation of the reliability when they examine the slope of eGFR decline. 

Response 1:  Thank you for pointing this out. We completely agree this comments. Therefore we revised the information. We described the enrollment criteria of the patients.  Patients with stages 3a-5 CKD and with renal anemia (plasma hemoglobin less than 13 and 12 g/dl in male and female, respectively) were enrolled (line 206-212). Patients taking steroid or immunosupressive drugs were excluded. Main causes of chronic renal failure were chronic glomerulonephritis, benign nephrosclerosis and diabetic nephropathy. Since daprodustat could be used starting 3 years ago in Japan, the treatment period was shorter than pre-period. Daprodustat is a drug to treat renal anemia, and it took time to consider the possibility that is a renoprotective drug.

 Comment 2:  More detailed information about the patients is needed. For example, what is the dosage and frequency of the daprodustat treatment of these patients? In addition to hypertension and hyperlipidemia, do they have any other health conditions that may potentially affect renal function, e.g. diabetes? Did they receive any other treatment such as dialysis? Their ethnicity? The authors may need to follow appropriate report guidelines, e.g. STROME. 

Response 2:  Thank you for pointing this out. We completely agree this comments. Therefore we revised the information. Thank you for the guideline, STROBE. We read and followed the guideline. Two and 4 mg of daprodustat were administrated daily to 19 and 4 patients, respectively (line 70-71). Therefore, the mean dose of 2.25 mg daprodustat was given to the patients. Diabetes was treated only in three patients. Patients who received renal replacement therapy were excluded (line 209).

Comment 3: The limitations of the study need to be discussed. For example, this study only enrolled 23 patients. The small sample size may limit the reliability of the study. 

Response 3:  Thank you for pointing this out. We completely agree this comments. Therefore we revised the information. Our examination is a retrospective study and the enrolled patients are a small number. We added this limitation of our study in the discussion section (line 198-200). We think that our manuscript will be the introduction to future prospective large scale studies.

Reviewer 2 Report

Comments and Suggestions for Authors

I read with interest the paper titled "Renoprotective effects of daprodustat in patients with chronic kidney diseases and renal anemia"

- Objective of the study should be stated in the abstract, as well as in the end of the introduction.

- Is this a cohort study? This is poorly described in the methods. Study design should be clearly stated, as well as recruitment criteria (inclusion and exclusion)

- Some variables are described as mean, others as median. Whats the rationale about this? 

- Have the normality of data been tested? How? If so, which test was used? If so, what's the cut p-value used to be considered normality?

- How authors calculated or obtained the sample size? 

- Whats the period of the study?

- "Gender" should be "sex". 

- In tables, names of variables should be replaced by the explained terms. For the readers, the variable names is not informative enough. 

- Results - In 1st paragraph of the results, authors compared other studies. Results section is usually to present your own results, and discussion should be used for the comparison with previous results from similar studies. 

- What the model building method used in the regression? Stepwise? Enter? Could you further explore. 

- Last paragraph of 4.3 statistics is a result. Should be provided as results. 

Author Response

Comment 1: Objective of the study should be stated in the abstract, as well as in the end of the introduction.

Response 1:   Thank you for pointing this out. We completely agree this comments. Therefore we revised the information.  We added the objective of this study in the abstract and introduction according to your suggestion (line 25-26). The purpose of our study was to examine whether daprodustat has renoprotective effects in the progression of chronic renal failure in patientswith stages 3a-5 CKD.

Comment 2:  Is this a cohort study? This is poorly described in the methods. Study design should be clearly stated, as well as recruitment criteria (inclusion and exclusion)

 Response 2:   Thank you for pointing this out. We completely agree this comments. Therefore we revised the information. Our study is not a cohort study. All patients received daprodustat and the decline slopes of eGFR were compared between pre- and post-treatments (line 232-234). Patients with stages 3a-5 CKD and with renal anemia (plasma hemoglobin less than 13 and 12 g/dl in male and female, respectively) were enrolled. Patients taking steroid or immunosupressive drugs were excluded (line 206212). Main causes of chronic renal failure were chronic glomerulonephritis, benign nephrosclerosis and diabetic nephropathy (line 212-214).

Comment 3: Some variables are described as mean, others as median. Whats the rationale about this? 

  Response 3:  Thank you for pointing this out. We completely agree this comments. Therefore we revised the information. Median was used only for age and serum creatinine, since serum creatinine varied from 0.79 to 7.11 mg/dl. Very high serum creatinine and age largely affect the mean values. So, we showed age and serum creatinine by the median value in the text (line 214-215 and kine 224, respectively). But we showed all values in Table 1 by mean ± SD.

Comment 4: Have the normality of data been tested? How? If so, which test was used? If so, what's the cut p-value used to be considered normality?

Response 4: Thank you for pointing this out. We made mistakes and completely agree this comments. Therefore we revised the information.  Pre- and post-hemoglobin were normally distributed (p=0.0529 and 0.5482 by Kolmogorov-Smirnov test, p=0.1605 and 0.9325 by Shapiro-Wilk test). Therefore, paired t-test was used to compare the difference of mean values (line 71-73). The slopes of pre- and post-eGFR, and pre- and post-U-TP/Ucr were not normally distributed. Therefore, the results by paired t-test were removed and the significance of the difference between two groups was examined by Wilcoxon Signed-rank test (line 77-79 and line 89-90).

Comment 5:  How authors calculated or obtained the sample size? 

Response 5:  Thank you for pointing this out. We completely agree this coments. Therefore we revised the information. We collected patients in Kitasato University Medical Center, Sagamihara Red-Cross Hospital and Kumamoto University Hospital.  Since daprodustat has been used for the treatment of renal anemia and not for the examination of renoprotection, the number of patients is quite low at present. The absence or presence of renoprotective effects of daprodustat or vadadustat was not examined in the large scle studies (ref. 15 and 16a9. The failure of showing renoprotetive effects by large scale studies with ESAs and the same stimulation of interstitial cells-derived Epo production both by ESAs and HIF-PHD inhibitors have counteracted the examination of the possibility of renoprotection by HIF-PHD inhibitors (line 51-55). It took time to consider the possibility of daprodustat as renoprotective drug

Comment 6: - Whats the period of the study?

Response 6:   Our retrospective study was performed from December 2023 to July 2024.

Comment 7:  "Gender" should be "sex". 

 Response 7:    Thank you for pointing this out. Gender was corrected to sex.

Comment 8:- In tables, names of variables should be replaced by the explained terms. For the readers, the variable names is not informative enough

Response 8:  Thank you for pointing this out.  We changed the explanation of variables in Tables.

Comment 9: Results - In 1st paragraph of the results, authors compared other studies. Results section is usually to present your own results, and discussion should be used for the comparison with previous results from similar studies. 

Response 9:  Thank you for pointing this out. We completely agree this comments. Therefore we revised the information.  We removed the description of the comparison with other studies (line 73).

Comment 10- What the model building method used in the regression? Stepwise? Enter? Could you further explore. 

 Response 10:  Thank you for pointing this out. We revised the information. Forward Stepwise Selection was used for the multiple regression analysis (line 142-143 and line 244-246).

Comment 11: Last paragraph of 4.3 statistics is a result. Should be provided as results. 

 Response 11: Thank you for pointing this out. We completely agree this comments. Therefore we revised the information. Part of the explanation was moved to the results according to your suggestions (line 248-252 and line 142-147).

Round 2

Reviewer 2 Report

Comments and Suggestions for Authors

The authors answered all the comments provided. Accept in the current form.